

# The HOSPITAL score as a predictor of 30 day readmission in a retrospective study at a university affiliated community hospital

Robert Robinson

Internal Medicine, Southern Illinois University School of Medicine, United States

Corresponding author
Robert Robinson,
rrobinson@siumed.edu

## ABSTRACT

**Introduction**. Hospital readmissions are common, expensive, and a key target of the Medicare Value Based Purchasing (VBP) program. Risk assessment tools have been developed to identify patients at high risk of hospital readmission so they can be targeted for interventions aimed at reducing the rate of readmission. One such tool is the HOSPITAL score that uses seven readily available clinical variables to predict the risk of readmission within 30 days of discharge. The HOSPITAL score has been internationally validated in large academic medical centers. This study aims to determine if the HOSPITAL score is similarly useful in a moderate sized university affiliated hospital in the midwestern United States.

**Materials and Methods**. All adult medical patients discharged from the SIU-SOM Hospitalist service from Memorial Medical Center (MMC) from October 15, 2015 to March 16, 2016, were studied retrospectively to determine if the HOSPITAL score was a significant predictor of hospital readmission within 30 days.

**Results**. During the study period, 998 discharges were recorded for the hospitalist service. The analysis includes data for the 931 discharges. Patients who died during the hospital stay, were transferred to another hospital, or left against medical advice were excluded. Of these patients, 109 (12%) were readmitted to the same hospital within 30 days. The patients who were readmitted were more likely to have a length of stay greater than or equal to 5 days (55% vs. 41%, $p = 0.005$) and were more likely to have been admitted more than once to the hospital within the last year (100% vs. 49%, $p < 0.001$). A receiver operating characteristic evaluation of the HOSPITAL score for this patient population shows a C statistic of 0.77 (95% CI [0.73–0.81]), indicating good discrimination for hospital readmission. The Brier score for the HOSPITAL score in this setting was 0.10, indicating good overall performance. The Hosmer–Lemeshow goodness of fit test shows a $\chi^2$ value of 1.63 with a $p$ value of 0.20.

**Discussion**. This single center retrospective study indicates that the HOSPITAL score has good discriminatory ability to predict hospital readmissions within 30 days for a medical hospitalist service at a university-affiliated hospital. This data for all causes of hospital readmission is comparable to the discriminatory ability of the HOSPITAL score in the international validation study (C statistics of 0.72 vs. 0.77) conducted at considerably larger hospitals (975 average beds vs. 507 at MMC) for potentially avoidable hospital readmissions.

**Conclusions**. The internationally validated HOSPITAL score may be a useful tool in moderate sized community hospitals to identify patients at high risk of hospital

readmission within 30 days. This easy to use scoring system using readily available data can be used as part of interventional strategies to reduce the rate of hospital readmission.

## INTRODUCTION

Hospital readmissions are common and expensive, with nearly 20% of Medicare patients being readmitted to a hospital within 30 days of discharge at an overall cost of nearly 20 billion USD per year (*Jencks, Williams & Coleman, 2009*). Because of this high frequency and cost, hospital readmissions within 30 days of discharge are a target for health care cost savings in the Medicare Value Based Purchasing (VBP) program. The VBP aims to incentivize hospitals and health systems to reduce readmissions through reductions in payments to hospitals with higher than expected readmission rates (*Centers for Medicare and Medicaid Services, 2016*). Because of the VBP initiative, health care organizations are investing considerable resources into efforts to reduce hospital readmission.

The underlying causes of hospital readmission are diverse. Studies have identified age, race, having a regular health care provider, major surgery, medical comorbidities, length of hospital stay, previous admissions in the last year, failure to transfer important information to the outpatient setting, discharging patients too soon, the number of medications at discharge, and many other risk factors for hospital readmission within 30 days (*Auerbach et al., 2016*; *Picker et al., 2015*; *Hasan et al., 2010*; *Silverstein et al., 2008*). Despite identifying with these risk factors, healthcare providers have poor accuracy in predicting which patients are at high risk of hospital readmission without a risk assessment tool (*Allaudeen et al., 2011*).

Readmission risk assessment can be accomplished with a variety of assessment tools that range from multidisciplinary patient interviews to simple screening tools using a handful of variables (*Zhou et al., 2016*; *Kansagara et al., 2011*; *Silverstein et al., 2008*; *Smith et al., 2000*). These tools use risk factors such as age, ethnicity, socioeconomic status, severity of illness, previous hospitalizations, and other factors to predict who is likely to be readmitted.

The easy to use HOSPITAL score is one such screening tool (*Donzé et al., 2013*). The HOSPITAL score uses seven readily available clinical predictors to accurately identify patients at high risk of potentially avoidable hospital readmission within 30 days. This score has been internationally validated in a population of over 100,000 patients at large academic medical centers (average size of 975 beds) and has been shown to have superior discriminative ability over other prediction tools (*Kansagara et al., 2011*; *Donzé et al., 2013*; *Donzé et al., 2016*).

This study aims to determine if the HOSPITAL score is a useful predictor of hospital readmission within 30 days of discharge in a moderate sized (507 bed) university affiliated hospital.

**Table 1  HOSPITAL score.**

| Attribute | Points if positive |
|---|---|
| Low hemoglobin at discharge (<12 g/dL) | 1 |
| Discharge from an Oncology service | 2 |
| Low sodium level at discharge (<135 mEq/L) | 1 |
| Procedure during hospital stay (ICD10 Coded) | 1 |
| Index admission type urgent or emergent | 1 |
| Number of hospital admissions during the previous year | |
|     0–1 | 0 |
|     2–5 | 2 |
|     >5 | 5 |
| Length of stay ≥ 5 days | 2 |

## MATERIALS AND METHODS

All adult medical patients discharged from the SIU-SOM Hospitalist service from Memorial Medical Center from October 15, 2015 to March 16, 2016, were studied retrospectively to determine if the HOSPITAL score was a significant predictor of any cause (avoidable and unavoidable) hospital readmission within 30 days.

Exclusion criteria were transfer to another acute care hospital, leaving the hospital against medical advice, or death.

The any cause readmission within 30 days of hospital discharge endpoint was selected because it is the measure used by the Medicare VBP.

Memorial Medical Center is a 507 bed not-for-profit university-affiliated tertiary care center located in Springfield, Illinois, USA. The SIU-SOM Hospitalist service is the general internal medicine residency teaching service staffed by board certified or board eligible hospitalist faculty. Patients for the hospitalist service are primarily admitted via the hospital emergency department or transferred from other regional hospitals with acute medical issues. Elective hospital admissions are extremely rare for this service.

Data on age, gender, diagnosis related group, length of stay, hospital readmission within 30 days, and the 7 variables in the HOSPITAL score (Table 1) were extracted from the electronic health record in a de-identified manner for analysis. Laboratory tests were infrequently obtained on the day of hospital discharge for hemoglobin (11%) and sodium (54%). Missing laboratory data (hemoglobin and sodium from the day of discharge) were coded to be in the normal range. Administrative approval for individual International Classification of Disease (ICD) code analysis was not granted for this study.

The study hospital does not have a distinct oncology admitting service. To partly address the increased risk of readmission in oncology patients found in other studies, this study classified patients with oncology related diagnosis related group (DRG) codes to have been discharged from an oncology service. This reflects local practice patterns where hospitalists often admit patients to the general medicine service for oncologists.

Because data is only available from the study hospital, readmissions at other hospitals will not be detected.

Institutional review board review for this study was obtained from the Springfield Committee for Research Involving Human Subjects. This study was determined not to meet the criteria for research involving human subjects according to 45 CFR 46.101 and 45 CFR 46.102.

### Statistical analysis

The HOSPITAL score was investigated as a predictor of any cause hospital readmission within 30 days. Qualitative variables were compared using Pearson $chi^2$ or Fisher's exact test and reported as frequency (%). Quantitative variables were compared using the non-parametric Mann–Whitney $U$ or Kruskal–Wallis tests and reported as mean $\pm$ standard deviation.

The HOSPITAL score was calculated for each admission. Scores of 0–4 points were classified as low risk for readmission (5%), 5–6 points intermediate risk (10%), and 7 or more points as high risk (20%) based on the initial validation study of the HOSPITAL score (*Donzé et al., 2013*). These readmission risk predictions were used to calculate a Brier score.

Most statistical analyses were performed using SPSS version 22 (SPSS Inc., Chicago, IL, USA).

The Brier score was calculated with R version 3.3.1 (R Foundation for Statistical Computing, Vienna, Austria).

Two sided $P$-values <0.05 were considered significant.

## RESULTS

During the study period, 998 discharges were recorded for the SIU-SOM Hospitalist service. The analysis includes data for the 931 discharges that met inclusion criteria (Fig. 1). Of these discharges, 109 (12%) were readmitted to the same hospital within 30 days. The study population was 52% female, had an average age of 63 years, and spent an average of 5.5 days in the hospital.

Discharged patients who were readmitted were more likely to have a length of stay greater than or equal to 5 days (55% vs. 41%, $p = 0.005$) and were more likely to have been admitted more than once to the hospital within the last year (100% vs. 49%, $p < 0.001$, Table 2). A receiver operating characteristic (ROC) evaluation of the HOSPITAL score for this population showed a C statistic of 0.77 (95% CI [0.73–0.81]) indicating good discrimination for hospital readmission (Fig. 2). The Brier score for the HOSPITAL score in this setting was 0.10, indicating good overall performance. The Hosmer–Lemeshow goodness of fit test showed a $\chi^2$ value of 1.63 with a $p$ value of 0.20.

## DISCUSSION

This single center retrospective study indicates that the HOSPITAL score has good discrimination and calibration to predict all cause hospital readmissions within 30 days for a medical hospitalist service at a university-affiliated hospital. The rate of readmission within 30 days in this population was 12%, which is less than the 20% rate of readmission seen in Medicare patients in a nationwide sample (*Jencks, Williams & Coleman, 2009*).

This data for all-causes of hospital readmission is comparable to the discriminatory ability of the HOSPITAL score in the international validation study (C statistics of 0.77

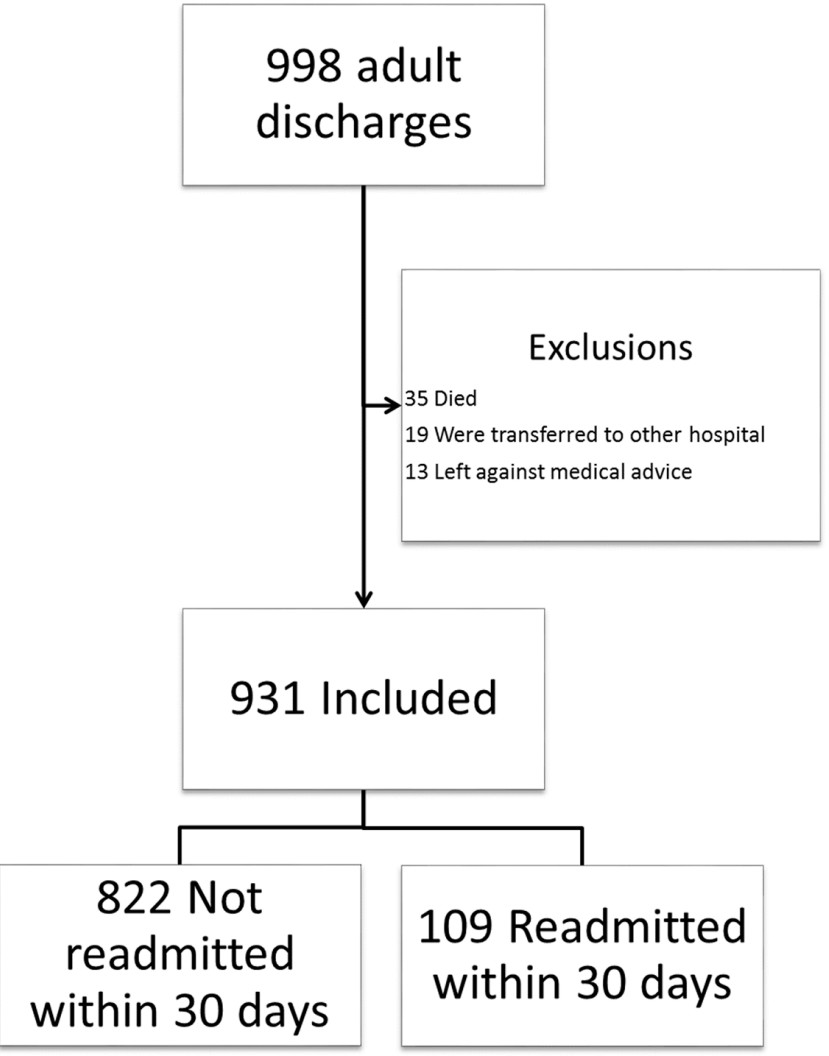

**Figure 1** **Study flow diagram.**

vs. 0.72) conducted at considerably larger hospitals (975 average beds vs. 507 at Memorial Medical Center) for potentially avoidable hospital readmissions (*Donzé et al., 2016*).

The HOSPITAL score had good overall performance in this setting with a Brier score of 0.10 and a Hosmer–Lemeshow goodness of fit test showing a $\chi^2$ value of 1.63 with a *p* value of 0.20. The Brier score from this study is similar to the score reported in the validation study (*Donzé et al., 2016*). The validation study had a superior goodness of fit test, likely reflecting the considerably larger sample size (*Donzé et al., 2016*).

The study population differs from the international validation study of the HOSPITAL score in several important ways. The study hospital does not have a distinct oncology admitting service, all of the admissions during this timeframe were classified as urgent or emergent, and discharge day laboratory tests for hemoglobin (11% vs. 94%) and sodium (54% vs. 97%) were less frequently performed (*Donzé et al., 2016*). The derivation and international validation studies accepted the last laboratory tests for hemoglobin and sodium as the

**Table 2  Baseline characteristics of the study population by 30 day readmission status.**

| Characteristic | Not readmitted within 30 days $n = 822$ | Readmitted within 30 days $n = 109$ | |
|---|---|---|---|
| Age, mean (SD) | 63 (17.2) | 64 (15.8) | $P = 0.27$ |
| Female | 421 (51%) | 61 (56%) | $P = 0.36$ |
| Urgent or emergent admission | 845 (100%) | 118 (100%) | |
| Discharge from oncology service | 21 (3%) | 4 (4%) | $P = 0.52$ |
| Length of stay $\geq$ 5 days | 336 (41%) | 60 (55%) | $P = 0.005$ |
| Hospital admissions in the last year | | | |
|     0–1 | 422 (51%) | 0 (0%) | $P < 0.001$ |
|     2–5 | 362 (44%) | 72 (66%) | |
|     >5 | 38 (5%) | 37 (34%) | |
| An ICD10 coded procedure during hospitalization | 378 (46%) | 52 (48%) | $P = 0.76$ |
| Low hemoglobin level at discharge (<12 g/dL) | 45 (5%) | 11 (10%) | $P = 0.43$ |
| Low sodium level at discharge (<135 mEq/L) | 231 (28%) | 33 (30%) | $P = 0.65$ |
| HOSPITAL Score > 5 (High Risk) | 318 (39%) | 81 (74%) | $P < 0.001$ |

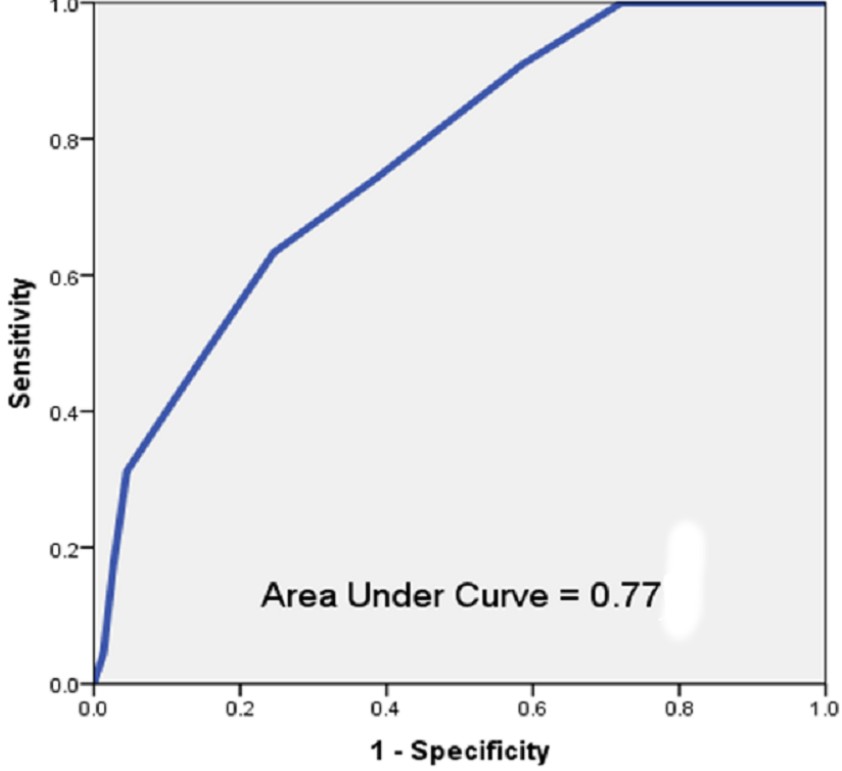

**Figure 2  Receiver operating characteristic curve of the HOSPITAL score in the study population.**

values at the time of discharge, this study only accepted results for tests performed on the day of discharge for these predictor variables (*Donzé et al., 2013*; *Donzé et al., 2016*). These factors are due to the local practice environment at the study site and are likely to have resulted in lower HOSPITAL scores for some discharges. This would lead to a reduced accuracy of the HOSPITAL score to predict readmission in this environment. Despite these differences, the HOSPITAL score performs well at this moderate sized university affiliated hospital.

The focus of all cause (avoidable and unavoidable) hospital admissions is a different endpoint than the potentially avoidable readmissions investigated in the derivation and validation studies for the HOSPITAL score (*Donzé et al., 2013*; *Donzé et al., 2016*). The endpoint of all cause readmissions is highly relevant because it is a significant marker of hospital quality under the Medicare VBP program for hospital reimbursement. Under this program, hospitals with high readmission rates can face financial penalties. To improve performance for this key healthcare quality measure, hospitals and health systems could use the HOSPITAL score to identify patients that may benefit from interventions directed at reducing hospital readmission. The HOSPITAL score is suitable for adaptation into an automated clinical decision support tool within an electronic health record system to identify patients at increased risk of hospital readmission.

This study has several important limitations. This study and the international validation study for the HOSPITAL score share an important shortfall by only identifying readmissions within 30 days at the same hospital (*Donzé et al., 2016*). Furthermore, this study is retrospective, single center, focused on medical patients, and shaped by local practice patterns (no oncology admitting service, few elective admissions, infrequent laboratory testing on the day of discharge). These limitations may reduce the generalizability of these results.

This study shows that the HOSPITAL score is useable in moderate sized community based hospitals to identify patients at high risk of readmission. Identifying these patients for interventions targeted at reducing hospital readmissions may result in improved patient care outcomes and healthcare quality.

## CONCLUSIONS

The internationally validated HOSPITAL score may be a useful tool in moderate sized community hospitals to identify patients at high risk of hospital readmission within 30 days. This easy to use scoring system using readily available data can identify patients at high risk for hospital readmission. These patients could then be targeted with interventional strategies designed to reduce the rate of hospital readmission.

Further research is needed to determine if the HOSPITAL score is a useful readmission risk prediction tool in other patient populations.

### Funding
The author received no funding for this work.

### Competing Interests
The author declares there are no competing interests.

## Author Contributions

- Robert Robinson conceived and designed the experiments, performed the experiments, analyzed the data, contributed reagents/materials/analysis tools, wrote the paper, prepared figures and/or tables, reviewed drafts of the paper.

## Human Ethics

The following information was supplied relating to ethical approvals (i.e., approving body and any reference numbers):

The Springfield Committee for research on human subjects reviewed the research proposal and determined that this project does not meet criteria for research involving human subjects according to 45 CFR 46.101 and 45 CFR 46.102.

## Data Availability

The raw data has been supplied as Data S1.

## Supplemental Information

Supplemental information for this article can be found online at http://dx.doi.org/10.7717/peerj.2441#supplemental-information.

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
