# Peer review of "The HOSPITAL score as a predictor of 30 day readmission in a retrospective study at a university affiliated community hospital"

_PeerJ, doi:10.7717/peerj.2441_

## Round 0.1 · original submission · Major Revisions

Reviewer 1 has made some minor suggestions that should be easy to adopt. Reviewer 2 has made detailed methodological and presentation remarks in the attached copy of your manuscript. Please review and respond to these critiques.

Reviewer 1 ·

Basic reporting

In this retrospective cohort study, the authors aimed to validate a prediction model (the HOSPITAL score) to identify patients at high risk for 30-day readmission. The populations studied is medical patients discharged from a moderate sized university affiliated hospital in the midwestern United States. In this population, the HOSPITAL score showed good performance. The article is well-written and methodology sounds.

My main constructive comments are as follows:

The article is very well written, clear, well-structured and using professional English language.
The study background and context is thoroughly described in the introduction, and adequately referenced. My only suggestion for the introduction would be to include the reference of the original study derivation of the HOSPITAL score directly after first mention on line 33 page of the introduction.
The figures and tables are relevant and of good quality.

Experimental design

The study is original, and bring valuable information for the practice.
The methods is well described, I would however suggest to mention the exclusion criteria (death) after line 49. I would also suggest to specify whether patients transferred to another hospital have been included in the analysis, or excluded as it was the case in the derivation study of the HOSPITAL score. As optional revision, authors could add the percentage of missing data.
The author should specify whether the outcome readmission included planned and unplanned readmission.
Because the hospital didn’t have a specific oncology unit, authors used DRG diagnosis of cancer instead. Please spell out DRG at first appearance. If the method sounds, one might wonder if the use of “any cancer diagnosis” based on the international classification of disease (ICD) codes instead may not be more accurate. Many cancer patients are hospitalized in an oncology unit because of infectious disease and would have as DRG a code for such diagnosis and not the cancer. It may explain the very low number of oncological patients presented in table 2 (only 26 patients out of the 963 included sounds few). Authors should consider if possible to add a sensitivity analysis using ICD codes instead of DRG codes.
Please add the number of patients categorized in the low and high risk group, along with their respective readmission rate.

Validity of the findings

Considering the rule of thumb of 10 outcomes per score variable in order to validate a prediction model, the sample size in this study sounds. Authors can consider to add this information. The data collected are appropriate for the study question.
Discussion is well-written and include the required information.

Reviewer 2 ·

Basic reporting

The article is clear and in professional English written.
Introduction and background show well the context, but literature is too sparse. The authors should add some additional relevant references.
The structure conforms to PeerJ standard.
For Figure 2, the authors should add the C-statistic in the labelling.
Raw data are supplied.

Experimental design

The article is within the scopes of PeerJ.
The research question is well defined, relevant and meaningful, and it is stated how it fills the knowledge gap which is identified.
Ethical issues are described.
The statistical analyses are however incomplete and I suggest some analyses to be added. The methods section describes insufficiently the analyses that were done.

Validity of the findings

Analyses are valid but incomplete.
Discussion and Conclusion are too short. They are not enough linked to original research question.

Additional comments

By investigating the validity of the HOSPITAL score to predict 30-days readmission in a moderate sized university affiliated hospital in the United States, Robinson studied an important topic. Hospital readmissions are indeed frequent and remain difficult to predict. The HOSPITAL score is one of the best tool to easily predict hospital readmissions, but had not yet been investigated in such a setting in the USA.
However, this manuscript need revision before being considered for publication in PeerJ, because of the issues mentioned in the attached file and in the above mentioned comments.

Annotated reviews are not available for download in order to protect the identity of reviewers who chose to remain anonymous.

---

## Round 0.2 · Minor Revisions

Both reviewers have agreed that your revisions have addressed the major issues associated with the initial submission. However, they have also addressed a few minor issues that should be addressed to improve the clarity and validity of the findings - please review and address these comments.

Reviewer 1 ·

Basic reporting

No comment

Experimental design

No comment

Validity of the findings

No comment

Additional comments

Reviewer comments have been addressed, thank you. Only one minor suggestion: the authors specified now that the lab values were collected at day of discharge only. However, in the original score derivation, the last available lab value (hemoglobin and sodium) was used, no matter when was made the blood draw. Therefore, a hemoglobin value on day 3 could be used as time of discharge at day 5 when no new blood draw has been performed. This may well explain the difference in the number of missing lab values between the current study and the derivation study. If the study data contain the last lab value as in the derivation study, I would recommend to run a sensitivity analysis. If not possible, please at least just add this difference in the type of predictor definition in the limitation section.

Reviewer 2 ·

Basic reporting

The language is clear. Some adaptations are needed to uniform the language throughout (see comments in the pdf). The structure conforms to PeerJ standard and Figures are ok. Raw data were supplied.

Experimental design

The subject fits the scopes of the journal and research question is very important. Methods is correctly described.

Validity of the findings

The findings are relevant and fill a gap in knowledge. Statistically sounds correct. Conclusion should be modified according to the original research question, because not limited to supporting the results (see comments in pdf).

Additional comments

Robinson et al. Studied the validity of a slightly adapted version of the HOSPITAL score in a moderate size university hospital. The modifications are relevant. The results fill a gap in knowledge and are well described. I just have some suggestions to modify before publication (see pdf attached).

Annotated reviews are not available for download in order to protect the identity of reviewers who chose to remain anonymous.

---

## Round 0.3 · accepted · Accept

Thank you for your careful consideration of the detailed comments from both reviewers.